# Double-Network Hydrogel for Stretchable Triboelectric Nanogenerator and Integrated Electroluminescent Skin with Self-Powered Rapid Visual Sensing

Yanshuo Sun [1,2,†], Jianjun Zhang [1,2,†], Chengyu Li [2,3,†], Jin Yang [1,2], Hao Li [1,2], Tao Jiang [1,2,3,*] and Baodong Chen [2,3,4,*]

1    School of Chemistry and Chemical Engineering, Guangxi University, Nanning 530004, China; sunyanshuo@binn.cas.cn (Y.S.); zhangjianjun@binn.cas.cn (J.Z.); yangjin@binn.cas.cn (J.Y.); lihao@binn.cas.cn (H.L.)
2    Beijing Institute of Nanoenergy and Nanosystems, Chinese Academy of Sciences, Beijing 101400, China; lichengyu@binn.cas.cn
3    School of Nanoscience and Technology, University of Chinese Academy of Sciences, Beijing 100049, China
4    Institute of Applied Nanotechnology, Jiaxing 314031, China
*    Correspondence: jiangtao@binn.cas.cn (T.J.); chenbaodong@binn.cas.cn (B.C.)
†    These authors contributed equally to this work.

**Abstract:** Bio-inspired design plays a very significant role in adapting biological models to technical applications of flexible electronics. The flexible, stretchable, and portable electrode is one of the key technical challenges in the field. Inspired by the responses to mechanical stimuli of natural plants, we designed a highly transparent (over 95%), stretchable (over 1500%), and biocompatible electrode material by using polymerized double-network hydrogel for fabricating a triboelectric nanogenerator (SH-TENG). The SH-TENG can convert tiny mechanical stretching from human movements directly into electrical energy, and is capable of lighting up to 50 LEDs. Benefiting from bio-inspired design, the coplanar, non-overlapping electrode structure breaks through the limitations of conventional electrodes in wearable devices and overcomes the bottleneck of transparent materials. Furthermore, a self-powered raindrop visual sensing system was realized, which can perform quasi-real-time rainfall information monitoring and increase rainfall recognition ability of vehicle automatic driving systems. This study provides a novel strategy for making next-generation stretchable electronic devices and flexible visual sensing systems.

**Keywords:** triboelectric nanogenerator; electroluminescent skin; self-powered visual sensing; double-network hydrogel

## 1. Introduction

Plant sensitivity to mechanical stimuli is enacted through adjustments of their vacuole morphology and stomatal structure, which can be initiated by wind, raindrops, and rubbing by passing animals. Bio-inspired design of plants' response to mechanical stimuli has long captured the curiosity of scientists and engineers, such as in bio-inspired materials, structures, and devices that have taken various forms. Bio-inspired design plays a very significant role as models for technical applications of flexible electronics; however, the development of a flexible, stretchable, and portable electrode is one of the key technical challenges in this field, and much plant-inspiration remains to be discovered and utilized. Stretchable, flexible, portable, and wearable electronic devices are significant in meeting the growing needs for the multi-functionality and sophistication of modern electronics, which have attracted widespread attention in the fields of big data, Internet of Things (IOT), and artificial intelligence [1–3]. However, traditional batteries are unable to satisfy the requirements of portable and wearable electronic devices for lightness and flexibility as a

result of their inherent defects such as bulkiness and stiffness. The triboelectric nanogenerator (TENG), based on the coupling of triboelectrification and electrostatic induction, is an attractive option for powering portable and wearable electronic devices [4,5]. It converts mechanical energy which is ubiquitous in the environment into electrical energy, and has emerged as a cutting-edge energy harvesting device, by virtue of its environmental friendliness and high output characteristics at low frequencies [6–8]. Moreover, the booming development of wearable electronic devices increasingly present miniaturization, portability, and low power consumption; this leads an inevitable trend to develop TENGs with high transparency and stretchability in order to improve the biocompatibility of the device and the comfort to the human wearer for human movement energy harvesting [9,10].

It is well known that the electrode plays a significant role in the integration of TENG into flexible and wearable applications [11]. Flexible, stretchable electrodes are typically fabricated using geometric design or filled elastomers with highly conductive materials, which entails drawbacks such as inhomogeneous mixing, poor electrical conductivity and weak adhesion [12]. In contrast, hydrogel possesses the advantages of high transparency, excellent stretchability, and superior biocompatibility, which make it an ideal material for wearable devices and flexible stretch sensors that are widely applied in tissue engineering, cartilage repair, drug carriers, smart sensors, and super-capacitors. In 2017, Xu [13] first reported a hydrogel-based self-powered human motion TENG sensor that promised huge potential in artificial skin, wearable electronic devices, and soft body robotics. Since then, the hydrogel-based TENG have gained widespread attention and research. For instance, Li's group [14] fabricated a composite hydrogel film using polyacrylamide (PAM)-combined $BaTiO_3$, which acts as electrodes for sensitive monitoring of human movement through piezoelectric, triboelectric, and piezoresistive effects. Yang et al. [15] reported a stretchable polyvinyl alcohol/phytic acid (PVA/PA) hydrogel-based TENG that possessed excellent flexibility and electrical properties for multi-channel signal acquisition and processing through bending fingers in human-machine interaction; it has a wide range of applications in intelligent medical systems. Guo et al. [16] designed an artificial haptic sensor via hydrogel to encode contact information into voltage pulses for the purpose of sensing approaching targets; it showed significant promise for applications in soft robotics, human-machine interaction, and intelligent prosthetics. Ionic hydrogels, which mainly rely on the migration of dissolved ions in the polymer network, are a growing field in flexible electronic devices. However, they are limited by their weak mechanical properties and poor tensile properties. Compared with other types of hydrogels, double-network hydrogels have aroused significant research interest by virtue of their biocompatibility, excellent mechanical strength, and unique energy dissipation mechanism.

Here, we propose simple and effective bio-inspired design ideas for fabricating electroluminescent skin based on the double-network hydrogels with high electrical conductivity, excellent stretchability (strain of 1540%), transparency (over 95%), and triboelectric properties; this skin enables us to produce the stretchable hydrogel-based triboelectric nanogenerator (SH-TENG). The SH-TENG is capable of lighting up to 50 LEDs directly by harvesting micro-nanomechanical energy from human movements. Owing to its excellent performance and improved structure, the SH-TENG is further utilized as an active conductive-type sensor for integration into a raindrop sensing windscreen. In order to provide high-sensitivity sensing and early warning, a self-powered raindrop visual sensing system was realized which can perform real-time rainfall monitoring for assisting vehicle automatic driving systems. This research opens new avenues for self-powered visual sensing in flexible and stretchable screens, intelligent display materials, and next-generation electronic skins.

## 2. Results and Discussion

### 2.1. Synthesis and Characterization of PAM-PVA Hydrogel

Benefiting from the reponse to mechanical stimulation of natural plants as shown in Figure 1a, a highly transparent (>95%), stretchable (>1500%), biocompatible double-

network conductive hydrogel material was fabricated using free-radical polymerization and hydrogen bonding; this acts as an electrode for coplanar electroluminescence and stretchable TENG in order to achieve raindrop luminescence as shown in Figure 1b,c; the SEM image of the PAM-PVA$_{10}$ hydrogel was taken in the inset. The freeze-dried hydrogels exhibited apparent three-dimensional network structure with regular distribution of macropores. Meanwhile, the double-network hydrogel was composed of two networks with chemical interactions, entanglements, and interactions. It is a tightly cross-linked, rigid, and brittle network, as well as a loosely cross-linked soft and tough network [17–19]. The PAM-PVA hydrogels were synthesized by the polymerization of acrylamide (AM), N, N′-methylenebisacrylamide (BIS), ammonium persulfate (APS), polyvinyl alcohol (PVA), and glycerol, which were employed as monomer, cross-linker, and initiator, respectively, as shown in Figure 1d. Firstly, the initial stage of chemical cross-linking of the PAM polymer cross-linking network was formed by the polymerization of AM monomer via a free-radical polymerization reaction. Meanwhile, the second stage physically cross-linked network structure formed interpenetrations with the PAM polymer network structure based on the hydrogen bonding between the amide groups in the PAM polymer chains with hydroxyl groups of PVA and glycerol molecules, resulting in a chemically and physically cross-linked PAM-PVA double network. In addition, hydrogels of different components can be prepared by simply adjusting the ratio of PVA. Figure 1e shows the results of investigating the effect of PVA on the PAM-PVA mixture using X-ray diffraction (XRD), in which a diffraction peak can be observed at about 23° in the X-ray diffraction curve, and this increases with an increase in PVA content. Meanwhile, the characteristic peaks of the prepared hydrogel lyophilized samples were characterized using FTIR. As shown in Figure 1f, the absorption peaks detected at 1500, 3000, and 3650 cm$^{-1}$ are the -C = O, -C = C, and -NH$_2$ stretching vibrations of AM, respectively. In addition, the SEM images of hydrogels with different PVA components are shown in the Supplementary Figure S2. The energy dispersive X-ray (EDX) spectrogram (Figure 1g) illustrates the homogeneous distribution of C, O, N, and P elements in the hydrogel.

### 2.2. Mechanical Properties of PAM-PVA Hydrogel

The double-network hydrogels were fabricated using a thermally initiated free-radical polymerization reaction and hydrogen bonding between macromolecules. In order to test its transparency, the transmittance of the hydrogel was measured in the wavelength range of 200–800 nm using a UV-Vis spectrometer. As shown in Figure 2a, the transparency of the prepared hydrogel was close to 100%. Meanwhile, the hydrogels which contained different contents of PVA were fabricated by simply regulating the content of PVA in order to further investigate the role of composition in mechanical properties, as well as to perform a series of tensile tests using a mechanical testing machine. Figure 2b illustrates that the hydrogels combined with PVA exhibit excellent mechanical properties compared to the pure PAM hydrogel. The maximum elongation at break reached 1540%. The tensile strength of 290 kPa was achieved with a PVA content of 10 mg/mL, which can be explained by the double-network structure. In particular, the chemically cross-linked PAM network structure imparts outstanding mechanical properties to the hydrogel, while the dynamically reversible physically cross-linked network structure imparts excellent elasticity and recovery ability. As the double-network ionic hydrogel was stretched, chemically cross-linked PAM acted as a sacrificial bond, which effectively dispersed the stress as it cracked into small clusters during the stretching process, hence realizing the tensile stress enhancement of the PAM-PVA hydrogel. The hydrogel has the greatest mechanical properties at a PVA content of 10 mg/mL; thus, we chose this hydrogel as the electrode and conducted the next test. However, the combination of PAM-PVA hydrogels soaked in lithium chloride (LiCl) produced a significant decrease in both tensile strength and strain, which was attributed to the weakening of the hydrogen bonds formed by the aggregation of conducting ions, as shown in Figure 2c. On the other hand, dissolved LiCl in water ionizes into Li$^+$ and Cl$^-$, which form hydrated ions with H$_2$O, resulting in an increase

in conductivity, with a maximum conductivity of about 0.3 S/m at the electrochemical workstation (Supplementary Figure S1b).

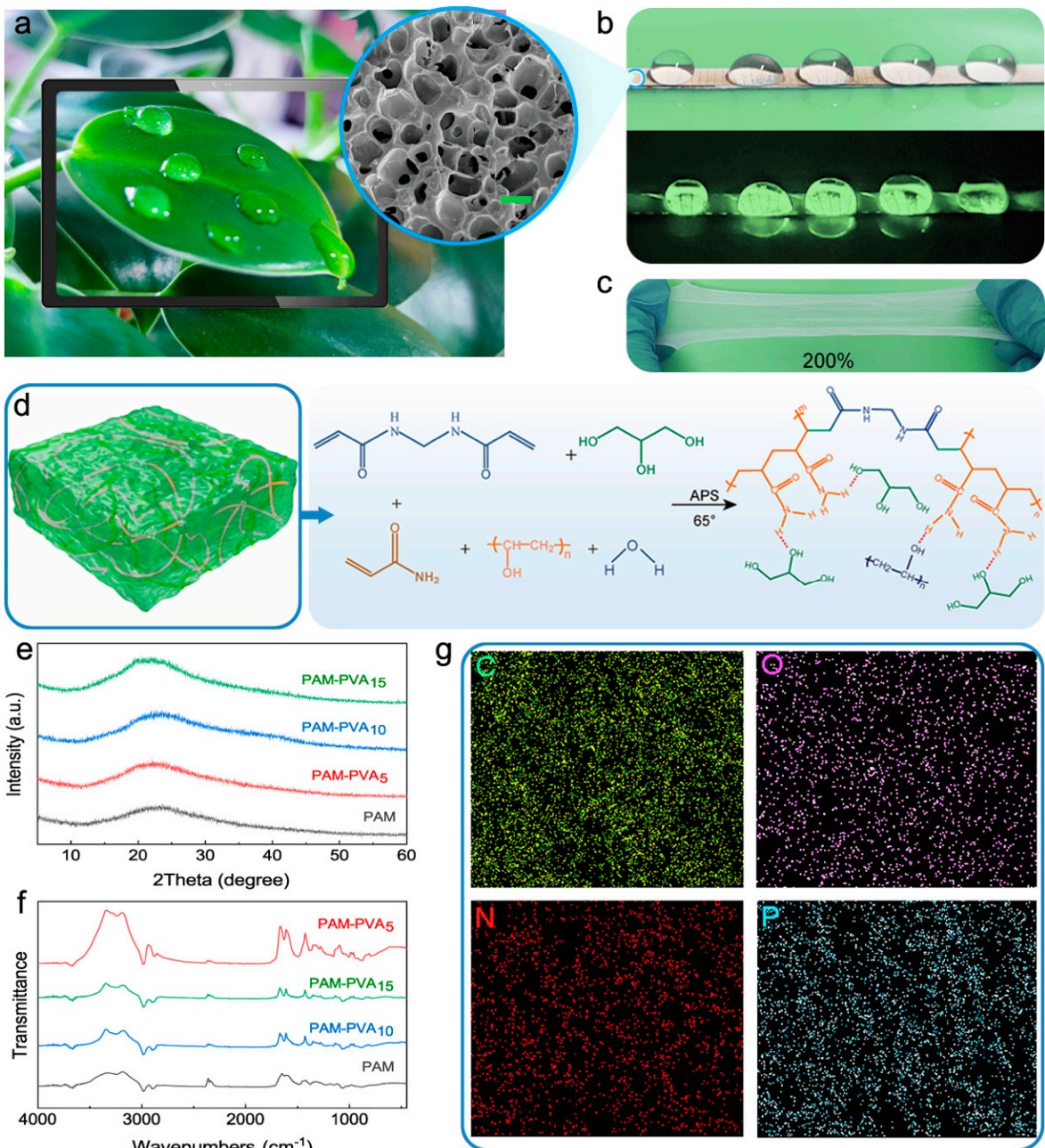

**Figure 1.** Schematic diagram and structural analysis of the PAM-PVA hydrogel. (**a**) The picture inspired by the response to mechanical stimuli of natural plants. (**b**) SEM of the hydrogel with a scale bar of 5 μm used for bio-inspired electroluminescent skin. (**c**) The hydrogel as an electrode for a stretchable triboelectric nanogenerator. (**d**) Schematic illustration of the synthesized PAM-PVA hydrogel showing the dual cross-link network. (**e**) XRD patterns of the hydrogel with different contents. (**f**) FTIR spectra of PAM, PAM-PVA$_5$, PAM-PVA$_{10}$, and PAM-PVA$_{15}$ hydrogels. (**g**) EDX maps of the PAM-PVA$_{10}$ hydrogel.

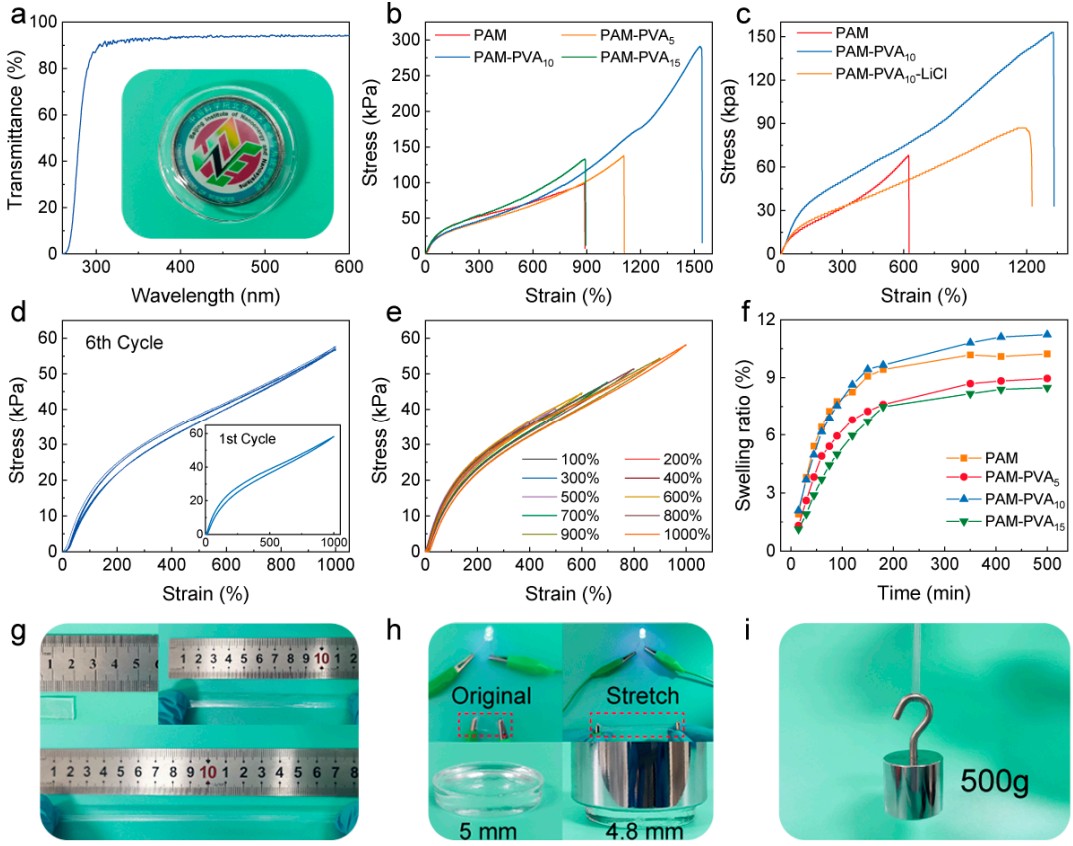

**Figure 2.** Tensile performances of the hydrogel with different polymer contents. (**a**) Transmittance curves of hydrogels with photos showing the transparency of the hydrogel. (**b**) Tensile curves of hydrogels with different components. (**c**) Tensile curves of hydrogels. (**d**) Loading–unloading cycles of PAM-PVA$_{10}$ hydrogels for six consecutive cycles at a strain value of 1000%. (**e**) Tensile loading–unloading curves for PAM-PVA$_{10}$ hydrogel at strain from 100–1000%. (**f**) Swelling ratio of the hydrogels as a function of time. (**g**) Photograph showing the high stretchability of the hydrogel. (**h**) Photographs of the conductivity and compressibility of hydrogels. (**i**) Photograph of PAM-PVA$_{10}$ ionic hydrogel with 500 g weight.

In order to evaluate the mechanical reliability of the PAM-PVA$_{10}$ hydrogel, cycle stretching and unloading was performed as shown in Figure 2d. Upon repeated loading and unloading up to 1000% strain, the tensile stress value demonstrated a slight plastic deformation in comparing the second to sixth cycles to the first cycle. Furthermore, the appearance of a hysteresis loop was observed as the applied strain was increased from 100% to 1000% in Figure 2e. The freeze-dried processed hydrogels (Figure 2f) reached swelling equilibrium after 8 h of reabsorption, which then recovered their original shape, indicating the structural completeness of the polymer matrix inside the hydrogel. Meanwhile, the effect of the drying environment on the water loss rate of the hydrogel was simulated in an oven at 60 °C, as shown in Supplementary Figure S1a. The results revealed the emergence of the Malines effect by displaying favorable energy dissipation, further revealing the potential tensile mechanical properties of PAM-PVA hydrogels. As shown in Figure 2g, the PAM-PVA$_{10}$ conductive hydrogel can exhibit favorable stretching properties. In order to verify the resistance change characteristics of the PAM-PVA$_{10}$ conductive hydrogel with the double-network interpenetrating structure during the stretching process, the hydrogel was connected to a power supply using an LED as the conductor. It can be seen that the LED was successfully lit, indicating the satisfactory conductivity of the hydrogel. As the hydrogel was continuously stretched, the brightness of the LED became slightly dimmer, as shown in Figure 2h. The main reason for the loss in brightness is that

as conductive hydrogels undergo deformation under stretching, the resistance becomes higher [20,21]. In order to further explore the compressive behavior of the hydrogel, weight forces were exerted in order to apply deformation to the PAM-PVA$_{10}$ hydrogel; there was no obvious compression strain observed (Figure 2h). Furthermore, the robust hydrogel can even endure a heavy weight of up to 500 g without obvious shape variation (Figure 2i). Moreover, the transparent hydrogel also demonstrates high adhesive properties (Figure S3), favorable conductive properties, and effectively converts the stretching action stimulus from the external environment into resistance change signals; this phenomenon has broad application prospects in the fields of electronic skin, health monitoring, and human motion sensing.

### 2.3. Electrical Output Performance of the SH-TENG

The double-network hydrogel as ionic conductor electrode was integrated with silicone to construct a transparent, stretchable, single-electrode mode SH-TENG (Figure 3a) which can be bent, twisted, and stretched, as displayed in Figure 3b,c. The detailed experimental operations are shown in the Experimental Section. Figure 3d illustrates the mechanism of SH-TENG operation [22]. As the latex contacts the Ecoflex layer, a positive charge generates at the latex layer interface in addition to a negative charge of the same quantity and opposite electrical properties at the Ecoflex interface. There is no electron flow in the external circuit as a result of the complete balance between the frictional charges. When the latex is separated from the silicone layer and moves upward, the conductive hydrogel generates positive charges in order to compensate for the negative charges on the surface of the Ecoflex layer. Electrons flow from the PAM-PVA$_{10}$ hydrogel to the copper wire during the above process, producing a measurable but temporary current signal. The whole process is reversed as the latex layer moves down and approaches the Ecoflex layer. The positive charge induced by the copper wire decreases, causing electrons to flow in the opposite direction. The SH-TENG sequentially generates an alternating current continuously by repeating the operation periodically. Figure 3e illustrates that the open-circuit voltage gradually increases with increasing contact forces in the range of 0.5–20 N.

The electrical output performance of the SH-TENG was measured at different frequencies from 0.5 to 2.5 Hz (Figure 3f–h). As a result of the excellent conductivity of the hydrogel-based TENG, the open-circuit voltage and short-circuit charge transfer were observed to be relatively stable at 70 V and 25 nC, respectively, by linear motor contact separation at the hydrogel contact area of $3 \times 2$ cm$^2$. Meanwhile, the short circuit current gradually increases up to 0.7 µA as the frequency increases. The output performance is improved slightly compared to the literature [21], which demonstrates the enormous potential of the TENG as an energy supplier. Meanwhile, the fabricated SH-TENG possesses high stretchability that enables stretching up to 300% of its original length, as shown in Supplementary Figure S4a. Figure 3i–k illustrates the open-circuit voltage, short-circuit current and short-circuit transfer charge decreases as the stretching rate increases, which is attributed to the change in resistance and the decrease in contact area that occur during stretching. Figure 3l illustrates the charging capability of the SH-TENG for different capacitors at a frequency of 1 Hz, which becomes progressively faster as the capacitance decreases. In addition, an external load with different resistances was connected to the SH-TENG in order to evaluate its output performance in an external circuit. Figure 3m demonstrates that the current in the SH-TENG reduces as the external resistance increases. The power density reaches up to 37 mW/m$^2$ at an external resistance of 2 GΩ. The power density of the SH-TENG was suitable for powering a variety of commercial devices and LED lamps. As shown in Supplemetary Video S1, the SH-TENG generates sufficiently high voltage to drive 50 LEDs directly. Meanwhile, the collected power can be stored in a capacitor or battery for later use through rectification, and maintains stable output performance after 5 days, as shown in Supplementary Figure S4b. In order to evaluate the mechanical stability of the SH-TENG, we tested the output performance of the SH-TENG after 10,000 vertical contact–separation continuous cycles at a frequency of 1 Hz. The output voltage changes

minimally (Figure 3n), which clearly illustrates the remarkable durability and practical application of the SH-TENG.

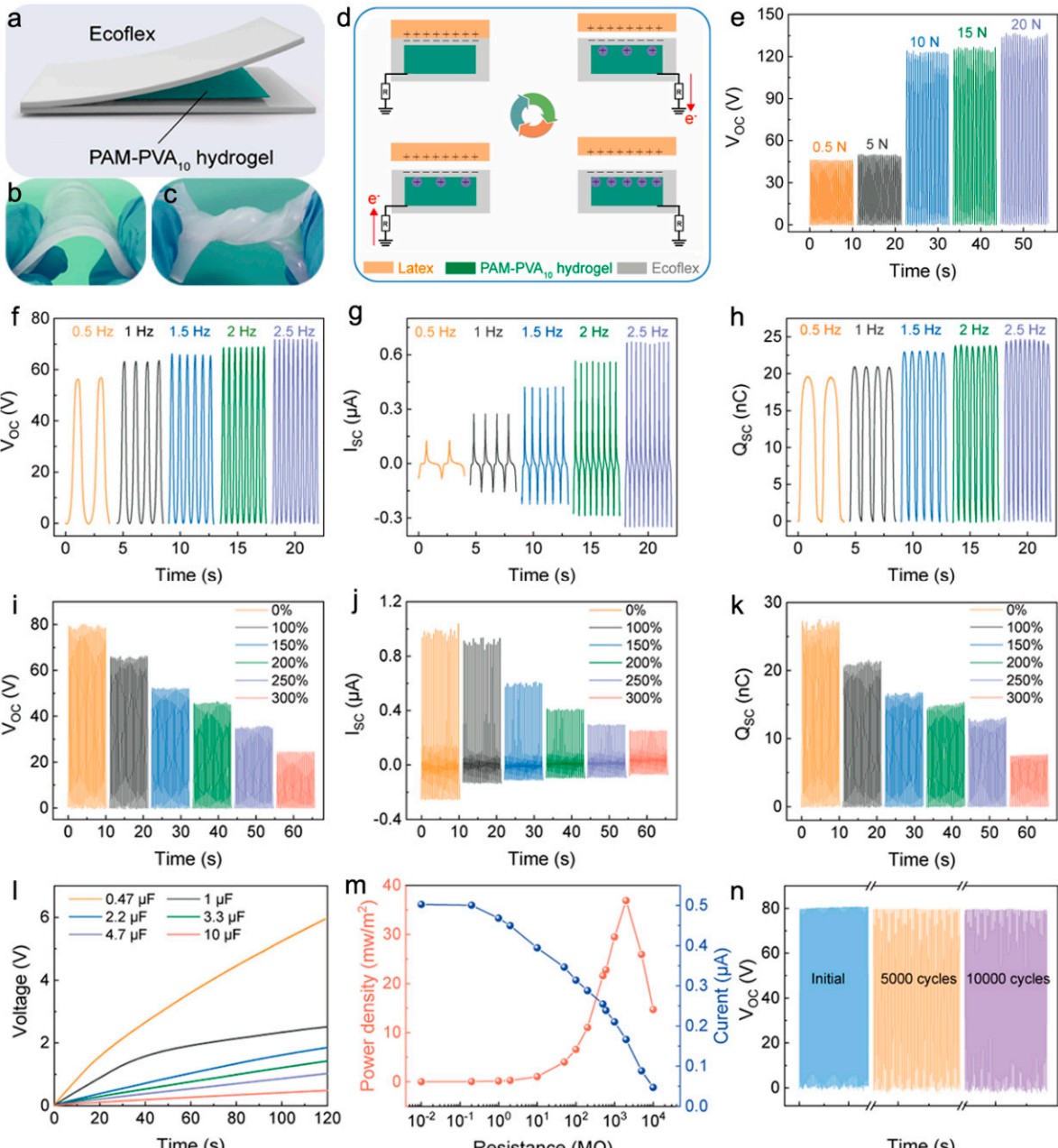

**Figure 3.** Performance of the SH-TENG based on PAM-PVA$_{10}$ hydrogel. (**a**) Schematic diagram of the SH-TENG based on PAM-PVA$_{10}$ hydrogel. (**b,c**) Photographs of the SH-TENG in original and twisted states. (**d**) Schematics of the operating principle for the SH-TENG. (**e**) Open-circuit voltages of the SH-TENG at various forces (0.5–20 N). (**f–h**) Open-circuit voltage, short-circuit current, and short-circuit transferred charges under different frequencies of the SH-TENG (0.5–2.5 Hz). (**i–k**) Output performance of the SH-TENG at different stretch rates. (**l**) Charging curves of the SH-TENG for different capacitors. (**m**) Output current and power density of the SH-TENG under various load resistances. (**n**) Output electrical stability test of the SH-TENG for 10,000 contact–separation cycles (at 1 Hz).

### 2.4. Bio-Inspired Electroluminescent Skin

Electroluminescent (EL) devices have drawn great research interest owing to their broad application in flexible lighting displays, electronic skin, and human-computer interactions [23–25]. The indium tin oxide (ITO) electrode is the most commonly used electrode for traditional alternating current electroluminescent devices; it possesses a number of remarkable properties, such as high transparency, conductivity, and low resistance; however, it is unable to stretch, and its rigidity limits wider applications of this electrode in flexible electronics, displays, and smart wearables. However, hydrogels have garnered the attention of many researchers thanks to their exceptional properties. Bio-inspired electroluminescent skins based on high stretchability, conductivity, and transparency have been fabricated, promoting the development of stretchable devices and intelligent wearables [26–28]. As shown in Figure 4a, the skin is primarily composed of electrodes, a phosphor layer, and an encapsulation layer. The luminescent layer was sandwiched between two symmetrical PAM-PVA$_{10}$ hydrogels and the dielectric silicone elastomer (Ecoflex 0030, purchased from Smooth-On Company, USA) layer as the encapsulation layer; its fabrication is shown in Section 4. Copper-doped zinc sulfide (ZnS:Cu) in the phosphor layer has excellent chemical stability and electroluminescence properties. In particular, ZnS as the primary material while Cu as the luminescent element in the center emits green or blue light upon excitation [29–33]. Emission materials can be easily tuned using doping with different concentrations and types of elements; incorporating and co-doping with different transition metals is an important way to prepare inorganic luminescent materials.

We investigated the luminescence performance of the bio-inspired electroluminescent skin under mechanical deformation. As shown in Figure 4b, the skin maintained light emission under conditions of stretching and bending, with light emission on both sides, which indicates that the skin was stable to mechanical deformation as a result of the elasticity and flexibility of the hydrogel. In terms of mechanical deformation stability, the prepared devices present brilliant emission with high homogeneity in different stretching states. As shown in the Figure 4c luminescence spectrum, the skin exhibits uniform and bright emission under a strain of 300%, where the central wavelength was 510 nm and the emission intensity increased as the tensile strain increased. The luminance intensity, as a function of tensile strain, is shown in Figure 4d. The photos show that the luminous intensity increases with increasing driving voltage. The main reason for this can be explained by the change in the thickness of the luminescent layer; the thickness of the luminescent layer gradually decreases as the stretching rate increases, which increases the electric field at an applied voltage, resulting in an increase in the brightness of the layer. The results verify the durability of stretchable electroluminescent skin to withstand a certain amount of mechanical deformation; this characteristic is critical for wearable applications. The conventional sandwich structure (Figure 4e) consists of a phosphor layer and dielectric layer sandwiched between a metal back electrode and a transparent electrically conductive front electrode, such as ITO. However, transparent electrodes are expensive and difficult to integrate into sensing modules directly, limiting their broader applications in material selection, smart wearables, and human-computer interactions. In contrast, a coplanar, non-overlapping electrode was proposed without optical transparency [34,35]. As shown in Figure 4f, light emission can be achieved by bridging electrodes (polar liquids such as water, hydrogel) on the top layer of the luminescent layer. Figure 4g displays a simple coplanar, non-overlapping electrode; it emits the bright green luminescence of a water droplet, which eliminates the limitation for transparent electrodes, and increases the prospect for a wider range of applications.

### 2.5. Self-Powered Raindrop Visual Sensing System

With the advantages of smart, safe, and precise positioning, car driving has become one of the most popular methods to travel; however, traffic accidents frequently occur as a result of obstructed vision in extreme weather conditions. Hence, it is proposed in this paper that a self-powered raindrop visual sensing system be used to realize real-time monitoring

of raindrops and rainfall levels. The application scenario of the raindrop sensing sensor is shown in Figure 5a. Figure 5b illustrates the structure of the raindrop sensing sensor, and Figure 5d shows a photograph of the physical setup, which mainly includes raindrop sensing and a luminous display. The sensors receive the signal, converted to voltage signals, as the raindrops fall, and an alarm is triggered when the minimum threshold is reached, in order to remind drivers of the arrival of rain. At the same time, raindrops as a bridge electrode fall on the coplanar electrode to achieve intelligent luminescence and alarm indication in the dark that significantly enhances the warning capabilities for night driving. As shown in Supplementary Video S2, the luminescence can also be realized during the stretching process, revealing excellent luminescence performance and stability, which can be widely applied in smart wear and stretch sensing in the future. Figure 5c and Supplementary Video S3 demonstrate the real-time monitoring process of the self-powered raindrop sensing system during an actual weather monitoring process; the system converts the raindrop signal into a real-time voltage signal through the signal acquisition and processing conversion system. Screenshots of the self-powered raindrop sensing system in real-time are shown in Figure 5e,f; the system attaches easily to the vehicle window, demonstrating the practical application feasibility and enormous potential for raindrop monitoring systems. Meanwhile, the voltage variation for a complete period, from raindrop impacting to dropping off, is shown in Figure 5g. The voltage signal increases sharply as the raindrops land, while the voltage decreases as the drops evaporate and disappear, returning to baseline value. Furthermore, the magnitude of the voltage is closely related to the amount of rainfall. Figure 5h illustrates the voltage output signal for different rainfall amounts, indicating that the voltage signal increases with the amount of rainfall, which has significant implications for future rainfall monitoring and intelligent flood control.

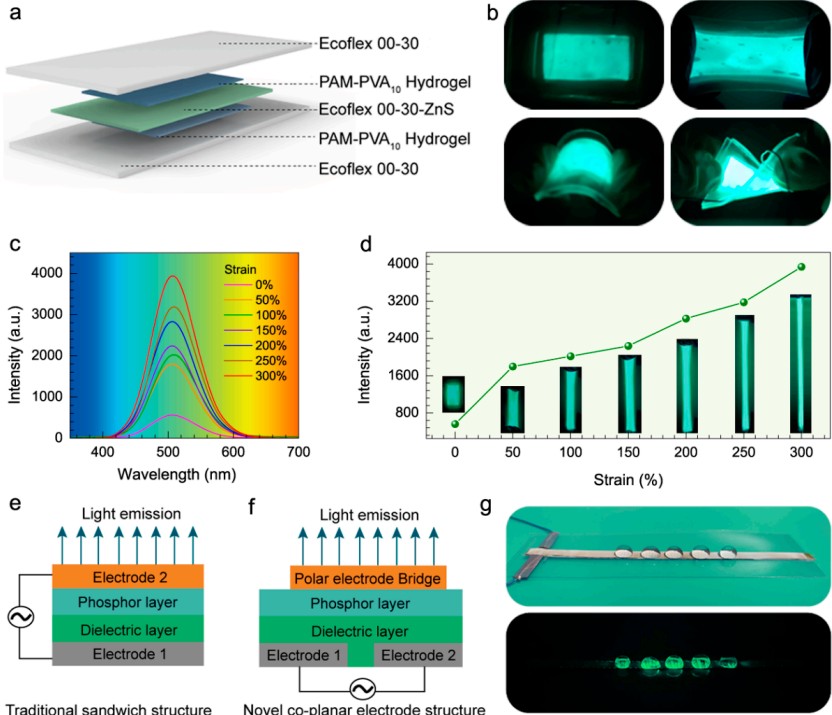

**Figure 4.** Structure and luminescent properties of bio-inspired electroluminescent skin. (**a**) Structural diagram of the stretchable electroluminescent skin device. (**b**) Photographs of the luminescence performance under bending and twisting deformations of the device. (**c**) Spectra under different stretching rates. (**d**) Corresponding spectrum intensity and optical photographs at different stretch rates. (**e**) Conventional sandwich structure. (**f**) Novel coplanar electrode structure. (**g**) Photograph of luminescence in water.

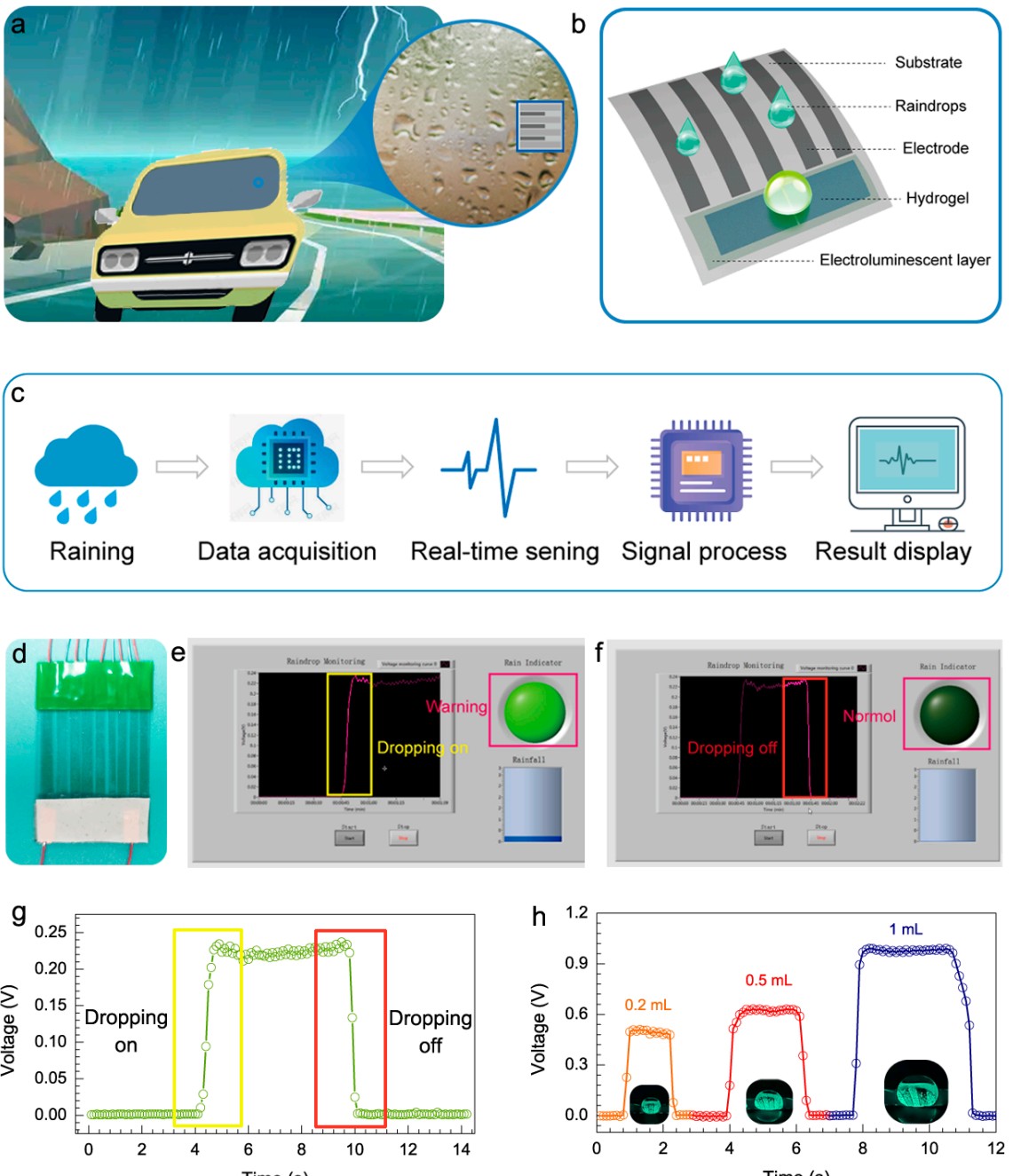

**Figure 5.** Demonstration of the developed self-powered raindrop visual sensing system. (**a**) Application scenario diagram for weather-monitoring sensors. (**b**) Schematic diagram of the structure of the sensor. (**c**) Real-time demonstration process in an actual application of the raindrop characteristic statistical system. (**d**) Physical view of the sensor. (**e**,**f**) Screenshot showing the self-powered raindrop characteristic statistical system monitoring changes in output voltage data. (**g**) Corresponding voltage data for falling raindrops. (**h**) Output voltage corresponding to various rainfall levels.

### 3. Conclusions

In summary, we developed a flexible, transparent, highly conductive, stretchable-hydrogel-based triboelectric nanogenerator (SH-TENG) for a self-powered visual sensing system. The SH-TENG can convert tiny mechanical stretching from human movements directly into electrical energy, and is capable of lighting up to 50 commercial LEDs. Benefiting from its bio-inspired design, the coplanar, non-overlapping structure of the electrode

surpasses the limitations of conventional electrodes in wearable devices, and overcomes the bottleneck of transparent materials. Owing to its excellent performance and bio-inspired design, the SH-TENG was further utilized as an active conduction type sensor for integration into a raindrop sensing windscreen. In order to provide high-sensitivity sensing and early warning, a self-powered raindrop visual sensing system was realized that can perform real-time rainfall monitoring to assisting vehicle automatic driving systems. This research is expected to lead to great opportunities in the development of electronic skins, and opens up new fields in stretchable hydrogel-based electronics combined with self-powered visual sensing.

## 4. Experimental Section

### 4.1. Materials

Polyvinyl alcohol (PVA) 1788, alcoholysis degree 87.0–89.0% (mol mol$^{-1}$), molecular weight 44.05, and N, N′-methylene bisacrylamide (BIS) were supplied by Shanghai Aladding Biochemical Technology Co., Ltd., Shanghai China. The copper-doped ZnS phosphor powder was synthesized by Shanghai Keyan Phosphor Technology Co., Ltd., Shanghai China. Ammonium persulfate (APS), acrylamide (AM), and glycerol were purchased from Shanghai Maclean Biochemical Co., Ltd., Shanghai China.

### 4.2. Synthesis of PAM-PVA Double-Network Hydrogels

PAM-PVA hydrogels with a double-network interpenetrating structure of chemically cross-linked network structure and physically cross-linked network structure were prepared by thermal initiation of a free-radical polymerization reaction and hydrogen bonding between macromolecules. First, 0, 100, 200, and 300 mg of PVA were dissolved in 20 mL of deionized water and heated in a water bath at 90 °C to obtain completely dissolved PVA solutions of different contents: 0, 5 mg/mL, 10 mg/mL, and 15 mg/mL, respectively. Then, the solutions were cooled to room temperature. PVA solution was transferred to a beaker, and 5 g AM, 0.005 g BIS, 120 μL glycerol, and 0.02 g APS were added to obtain a clear and transparent hydrogel pre-polymerization solution until the solutes dissolved completely. A certain volume of hydrogel pre-polymerization solution was poured into a petri dish and heated at 65 °C for 1 h in order to trigger the free-radical polymerization reaction, and obtain PAM-PVA conductive hydrogel with a double-network interpenetration structure.

### 4.3. Fabrication of the SH-TENG

The SH-TENG was a sandwich structure with Ecoflex as the top and bottom layer, with the hydrogel encapsulated inside. Ecoflex 00–30 silicone rubber was mixed in a ratio of 1:1 and transferred into the model with vacuum-suction in order to remove air bubbles; it was cured at room temperature for 3 h to fabricate the base layer. Next, the prepared PAM-PVA$_{10}$ hydrogel was placed in the center of the silicone rubber and attached to a conductive copper foil as the external electrode. Then, the same Ecoflex 00–30 was poured into the model as the top layer, followed by vacuum and curing. As the PAM-PVA hydrogel is encapsulated in two layers of silicone, it can effectively prevent the problem of water loss.

### 4.4. Fabrication of Self-Powered Raindrop Visual Sensing System

The system consists of five consecutive stacked layers. First, the prepared luminescent layer was sandwiched between the top and bottom ionic gel layers, and the Ecoflex elastomer was mixed with ZnS:Cu in a weight ratio of 1:1.5 and cured at 80 °C for 15 min. Both the top and bottom PAM-PVA$_{10}$ ionic gels were connected with copper wires to an external power supply, while the entire device was encapsulated with Ecoflex. The Ecoflex layer was prepared by mixing Ecoflex 00–30A and Ecoflex 00–30B in a 1:1 weight ratio.

### 4.5. Characterization of Materials and Devices

The XRD patterns were measured with powder XRD (PANalytical B. V., Xpert3 Power). The FTIR spectra were obtained by spectrometer (Bruker, Vertex80V). The micromorphology

of PAM-PVA hydrogel was tested using a Nova filed emission scanning electron microscope. The stress–strain of PAM-PVA hydrogels was tested using a YL-S71 Tensile testing machine, and all tensile rates were fixed at 80 mm-min$^{-1}$ unless specified otherwise. The bio-inspired electroluminescent skin spectrum was tested through a spectrometer. The UV−vis−NIR spectra for hydrogels were studied using a Shimadzu UV-3600 spectrophotometer at room temperature. A linear motor (LinMot E1100) was used to provide the driving force for the contact separation process of the SH-TENG. The open-circuit voltage, short-circuit current, and transferred charge amount of the SH-TENG were detected and recorded by a Keithley 6514 electrometer.

**Supplementary Materials:** The following are available online at https://www.mdpi.com/article/10.3390/electronics11131928/s1, Figure S1: Characterization of the double-network hydrogel properties, Figure S2: Scanning electron microscopy (SEM) images of the double-network hydrogel, Figure S3: Adhesive properties of PAM-PVA10 hydrogel, Figure S4: The output performance of the SH-TENG, Figure S5: The screenshots and photos of the self-powered raindrops visual sensing system, Figure S6: The screenshots of the raindrop monitoring, Video S1: SH-TENG drives 50 LEDs, Video S2: Hydrogel as electrode to achieve stretchable luminescence, Video S3: The system monitors the raindrop and luminous display.

**Author Contributions:** Y.S., J.Z. and C.L. contributed equally to this research. T.J., B.C. and Y.S. conceived the project and designed the experiments. Y.S., J.Z. and C.L. contributed to sample preparation. Y.S., J.Z. and C.L. performed the experiments. J.Y. and H.L. contributed to data analysis. All authors discussed the results and commented on the manuscript. T.J. and B.C. wrote the paper with input from all authors. All authors have read and agreed to the published version of the manuscript.

**Funding:** This research was funded by the National Natural Science Foundation of China (Grant No. 52192610), and National Key R & D Project from the Minister of Science and Technology (2021YFA1201601).

**Data Availability Statement:** The data presented in this study are available on reasonable request from the corresponding author.

**Acknowledgments:** The authors acknowledge the support from the National Natural Science Foundation of China (Grant No. 52192610), and National Key R & D Project from the Minister of Science and Technology (2021YFA1201601).

**Conflicts of Interest:** The authors declare that they have no known competing financial interests or personal relationships that could have appeared to influence the research reported in this paper.

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
