# Peer review of "Double-Network Hydrogel for Stretchable Triboelectric Nanogenerator and Integrated Electroluminescent Skin with Self-Powered Rapid Visual Sensing"

_electronics, doi:10.3390/electronics11131928_

Round 1
Reviewer 1 Report
The paper uses a bioinspired double-network hydrogels for TENGs and raindrop sensing system. Methods are clear and results look promising. Author should improve introduction and results section and proofread the whole article to correct grammar issues.
Some specific comments are:
1. Introduction
Consider rewrite literature review, it should include more literature studies and recent results on hydrogel-based TENGs and emphasize more on the novelty of the current study.
2. Results and discussion
Synthesis and characterization of PAM-PVA hydrogel
Figure 1b, please add scale for the SEM so that readers have an idea how big are the pores.
What does 5, 10,15 in PAM-PVA5, PAM-PVA10, PAM-PVA15 represent? Please clarify.
Mechanical properties of PAM-PVA hydrogel:
Explanation: why PAM-PVA 10 performed better than PAM-PVA5 and PAM-PVA15, why PAM-PVA 15 performs worse than PAM-PAV5. Please explain.
Electrical output performance of the SH-TENG:
The author should compare and analyze the experimental results with literature values for the similar type of hydrogel-TENGs.
Bioinspired electroluminescent skin:
Figure 4, should be (g) instead of (h)
3. conclusion
Page 6 line 286-289, as well as page 1 line 19-21check the grammar.
5. Associated content
Page 7 line 346, unfinished sentence.
Reviewer 2 Report
Overall, the paper present a nice idea. Inspired by mechanical stimuli of the natural plants, the manuscript proposed a transparent, flexible, stretchable PAM-PVA hydrogel, which has a broad application in TENG and ACEL devices, showing the self-powered raindrops monitoring and visual sensing. The material of designed and applications in this work are interesting and the reviewer believe it can be published in the journal of Electronics. Below are some suggestions for further improving the results.
1. The output performance of the TENG is a key indicator of superiority. However, why the output performance decrease as the strain ratio increases in the Fig.3 (i-k)?
2. Can the authors explain the advantages of the self-powered raindrops visual sensing system compare to other raindrops sensors?
3. The ACEL device exhibits excellent luminous properties, but not sure whether the luminescence intensity is high enough to be visible in natural daylight.
4. The authors are suggested to check the typos and spelling of some words throughout the manuscript.
